# Extracts of *Sideritis scardica* and *Clinopodium vulgare* Alleviate Cognitive Impairments in Scopolamine-Induced Rat Dementia

**DOI:** 10.3390/ijms25031840

**Published:** 2024-02-02

**Authors:** Maria Lazarova, Elina Tsvetanova, Almira Georgieva, Miroslava Stefanova, Diamara Uzunova, Petko Denev, Valya Vassileva, Krasimira Tasheva

**Affiliations:** 1Institute of Neurobiology, Bulgarian Academy of Science, 1113 Sofia, Bulgaria; m.lazarova@gmail.com (M.L.); elina_nesta@abv.bg (E.T.); almirageorgieva@gmail.com (A.G.); mira_stefanova@mail.bg (M.S.); didi_uzunova1@abv.bg (D.U.); 2Laboratory of Biologically Active Substances, Institute of Organic Chemistry with Centre of Phytochemistry, Bulgarian Academy of Sciences, 4000 Plovdiv, Bulgaria; 3Institute of Plant Physiology and Genetics, Bulgarian Academy of Sciences, Acad. Georgi Bonchev Str., Block 21, 1113 Sofia, Bulgaria; valyavassileva@bio21.bas.bg

**Keywords:** recognition memory, antioxidant system, acetylcholinesterase, biogenic amines, BDNF, pCREB

## Abstract

*Sideritis scardica* Griseb. and *Clinopodium vulgare* L., belonging to the Lamiaceae family, are rich in terpenoids and phenolics and exhibit various pharmacological effects, including antioxidant, anti-inflammatory and anti-cancer activities. While the memory-enhancing impacts of *S. scardica* are well documented, the cognitive benefits of *C. vulgare* remain unexplored. This study assessed the potential effect of *C. vulgare* on learning and memory in healthy and scopolamine (Sco)-induced memory-impaired male Wistar rats, comparing it with the effects of *S. scardica*. Over a 21-day period, rats orally received extracts of cultivated *S. scardica* (200 mg/kg) and *C. vulgare* (100 mg/kg), either individually or in combination, with administration starting 10 days before and continuing 11 days simultaneously with Sco injection at a dose of 2 mg/kg intraperitoneally. The results showed that both extracts effectively mitigated Sco-induced memory impairment. Their combination significantly improved recognition memory and maintained monoaminergic function. *S. scardica* excelled in preserving spatial working memory, while *C. vulgare* exhibited comparable retention of recognition memory, robust antioxidant activity and acetylcholinesterase inhibitory activity. The extracts alleviated Sco-induced downregulation of p-CREB/BDNF signaling, suggesting neuroprotective mechanisms. The extract combination positively affected most of the Sco-induced impairments, underscoring the potential for further investigation of these extracts for therapeutic development.

## 1. Introduction

Alzheimer disease (AD) is one of the most widespread age-related progressive neurodegenerative diseases [1]. Over the years, this disease gradually impairs memory, affects language and leads to abnormalities in the personality and behavior of patients [2]. The neurodegenerative process damages cognitive function in several ways, with a primary cause being the compromised functionality of the brain cholinergic system. This deficiency is characterized by a substantial loss of cholinergic neurons in the basal forebrain along with their projections to the cortex and hippocampus—integral brain structures essential for the maintenance and regulation of memory [3,4,5,6,7,8]. Additionally, decreased levels of acetylcholine (ACh) and choline acetyltransferase (CAT), coupled with elevated acetylcholinesterase (AChE) in the brains of dementia patients, have been well established [9,10]. AChE terminates Ach before it reaches the ACh receptor, hydrolyzing it to choline and acetic acid [11]. Therefore, the use of acetylcholinesterase inhibitors (AChEIs) is the most popular pharmacological strategy for the symptomatic treatment of AD [12].

Progressive neuronal atrophy in AD is associated with downregulation in the levels of Brain-Derived Neurotrophic Factor (BDNF)/cAMP response element binding protein (CREB) signaling [13,14,15]. This signaling cascade is closely related to synaptic plasticity and memory processes in both the hippocampus and cortex [16]. BDNF and its major receptor tropomyosin receptor kinase B (TrkB) are essential for the proper development and functioning of dopaminergic (DAergic), gamma-aminobutyric acidergic (GABA), cholinergic (ACh) and serotoninergic (Sero) neurons [17]. They participate in long-term memory formation in the hippocampus through the activation of the transcription factor CREB [18]. CREB regulates the downstream expression of target genes, including BDNF and B-cell lymphoma 2 (Bcl2), contributing to processes like neuronal plasticity and neurodegeneration [19].

The disease affects the brain monoaminergic system, worsening deficits. Extensive evidence reveals neuronal loss and reduced levels of neurotransmitters, receptors and metabolites in the DAergic, noradrenergic (NAergic) and serotonergic (Seroergic) systems in AD patients [20,21,22,23,24,25,26,27,28,29,30,31,32]. This dysfunction is primarily linked to neuropsychiatric symptoms in AD, including depression, agitation, anxiety, apathy, psychosis and disturbances in sleep or appetite in later stages [33,34]. DA is crucial for memory encoding in the hippocampus, acting as a modulator of hippocampal synaptic plasticity [35,36,37]. NAergic transmission affects the management of processes, such as synaptic plasticity, neuronal metabolism and permeability of the blood–brain barrier (BBB). Enhancing brain noradrenergic neurotransmission has been effective in reducing neuroinflammation and cognitive decline [38].

Current AD treatments have yielded limited results due to their narrow focus on specific nervous tissue pathophysiologies like cholinergic abnormalities or amyloid accumulation [39]. A broader approach is necessary to tackle the disease complexity. Traditional medications frequently lead to side effects, including hepatotoxicity, and the efficacy of new drugs remains to be proven [40,41]. Consequently, there is a continued search for alternative approaches, particularly natural compounds with multifaceted actions. Plants, with their diverse secondary metabolites, are excellent candidates for deriving such therapeutics.

*Sideritis scardica* Griseb. (mountain tea) and *Clinopodium vulgare* L. (wild basil) belong to the Lamiaceae family and have a longstanding presence in folk medicine. *S. scardica* is endemic to the Balkan Peninsula, and in Bulgaria the species is classified as critically endangered and included in the Red Book. Rich in polyphenols, hydroxycinnamic acid derivatives and phenylethanoid glycosides, it exhibits various physiological properties, including antioxidant effects [42,43,44] and anti-inflammatory [45] and antitumor [45,46,47,48] activities.

The therapeutic potential of *S. scardica* in neurodegenerative diseases, including AD, has been previously explored. These studies have shown a positive impact on cognition and memory in AD-amyloidosis mouse models and aged C57Bl/6 mice using *Sideritis euboea* and *S. scardica* extracts [49]. Clinical trials with healthy individuals under stress and elderly individuals with mild cognitive impairment also reported cognitive benefits from *S. scardica* [50,51,52]. Our data revealed that water extract from cultivated *S. scardica* aided memory formation in scopolamine (Sco)-induced dementia in mice, improving spatial working memory and reducing anxiety-like behavior [53]. We also noted a noradrenaline-preserving effect and serotonergic stimulation in this dementia model [53]. These results are consistent with in vitro findings by Knörle et al. [54], who reported *S. scardica* extracts acting as triple monoamine reuptake inhibitors in mouse brain synaptosomes [54] with antidepressant and psychostimulating effects in rats [52,55].

Wild basil is widely distributed in southern and southeastern Europe, North America, Latin America and Asia [56]. The phytochemical profile of the plant reveals the presence of phenolic acids, flavonoids, phenylpropanoids, caffeic acid oligomers and saponins, which are well known for their antioxidant and anti-inflammatory potential [57,58,59,60]. Existing data indicate that *C. vulgare* extract modulates cyclooxygenase-2 expression in neutrophils, impacting the inflammatory process. Active constituents such as catechin, caffeic and chlorogenic acids have been identified [61].

To the best of our knowledge, the effects of *C. vulgare* extracts on cognitive function in vivo have not been investigated hitherto. In a recent study, Bektašević et al. [62] reported weak AChE inhibitory activity of *C. vulgare* extract. Based on this finding and considering the ability of *C. vulgare* to suppress inflammation, a key factor in AD etiology, we hypothesize that it may also exhibit a neuroprotective effect. Furthermore, we speculate that *C. vulgare* may strengthen the cognitive-enhancing effect of *S. scardica* found in our previous studies conducted in mice with the Sco model of dementia.

The present study was designed to Investigate the memory-enhancing effect of water extract from cultivated *C. vulgare* and compare it with that of *S. scardica*, both individually and in combination using Sco-induced experimental dementia in rats. Cognitive changes in animal behavior were tested by T-maze and Novel Object Recognition (NOR) tests. To understand the molecular mechanisms underlying potential memory protection, aChE and antioxidant properties were assessed in the frontal cortex and hippocampus of rats. Moreover, the potential impact of the extracts on monoaminergic neurotransmitter systems in the brain and the expression levels of memory-related proteins, including phosphorylated cAMP response element-binding (CREB) and Brain-Derived Neurotrophic Factor (BDNF), were additionally examined.

## 2. Results

### 2.1. Total Polyphenol and Flavonoid Content and Antioxidant Activity of Plant Extracts

Polyphenols, particularly flavonoids, derived from the secondary metabolism of plants, have the capacity to suppress the spontaneous autoxidation of organic molecules. One of the antioxidant mechanisms of flavonoids involves suppressing the formation of reactive oxygen species (ROS) by inhibiting enzymes or chelating trace elements associated with the production of free radicals.

Table 1 displays the polyphenol and flavonoid content and the in vitro antioxidant activity of extracts from *C. vulgare*, *S. scardica* and their combination. Our results showed that the *C. vulgare* extract possessed the highest polyphenol content (22,402 ± 812 mg/100 g), followed by the extract combination (15,064 ± 307 mg/100 g) and *S. scardica* (12,096 ± 1208 mg/100 g). Regarding flavonoid content, *C. vulgare* ranked the highest (3689 ± 190 mg/100 g), followed by the combination (2755 ± 102 mg/100 g) and *S. scardica* (1903 ± 229 mg/100 g). In terms of in vitro antioxidant activity, assessed through oxygen radical absorbance capacity (ORAC) and hydroxyl radical averting capacity (HORAC) assays, the *C. vulgare* extract showed the highest antioxidant properties.

### 2.2. Effects of Plant Extracts on Rewarded Spontaneous Alternation Behavior in T-Maze Test

The T-maze test served as a tool to evaluate changes in the spatial working memory of the animals in different experimental groups. The analysis of cognitive function in rats was conducted 12 days after the first Sco treatment (Figure 1). Our results showed that in the Sco-treated group, the number of correct choices was significantly lower (by 32.56%, *p* < 0.05, *n* = 10) compared to the control group, indicating memory impairment. However, the Sco-induced decrease in cognitive function was significantly reversed by the *S. scardica* plant extract after 21 days of treatment. The number of correct choices during the test increased by 37.98% (*p* < 0.001, *n* = 10) compared to the untreated Sco group.

In groups of healthy rats treated with plant extracts, the memory performance of the animals did not significantly change compared to the control group.

### 2.3. Effects of Plant Extracts on Recognition Memory in NOR Test

An analysis of recognition memory using the discrimination index (DI) revealed that in the Sco-treated group, the animals spent less time with the novel object, indicating a memory decline (Figure 2). The DI in this group decreased by 68.65% (*p* < 0.001, *n* = 8) compared to the control. However, after 21 days of treatment, the plant extracts significantly reversed the Sco-induced memory impairment. Administration of *S. scardica*, *C. vulgare* and the combination of *S. scardica* + *C. vulgare* increased DI by 44.24% (*p* < 0.05, *n* = 8), 44.04% (*p* < 0.05, *n* = 8) and 58.53% (*p* < 0.001, *n* = 8), respectively, as compared to the Sco group. In healthy animals treated with various plant extracts, the DI was not significantly changed compared to the control group.

### 2.4. Effects of Plant Extracts on Oxidative Stress Parameters in the Frontal Cortex and Hippocampus of Healthy and Scopolamine-Treated Rats

The effect of plant extracts derived from *S. scardica*, *C. vulgare* and their combination on non-enzyme components of the oxidative stress defense system in cells was analyzed. These effects were assessed in the cortex and hippocampus of both healthy and demented rodents.

#### 2.4.1. Effects on MDA Level

As depicted in Figure 3A,B, the 11-day treatment with Sco significantly increased malondialdehyde (MDA) levels in the cortex (by 22.20%, *p* < 0.01, *n* = 6) and hippocampus (by 32.77%, *p* < 0.001, *n* = 6) of the rodents compared to the control, indicating escalated oxidative stress. In the cortex, the Sco-induced changes in MDA levels were prevented by *C. vulgare* (by 49.28%, *p* < 0.001, *n* = 6) and the combination of *S. scardica* + *C. vulgare* treatment (by 26.89%, *p* < 0.001, *n* = 6). In the hippocampus, *S. scardica* and *C. vulgare* demonstrated antioxidant capacity, reducing MDA levels by 11.94% (*p* < 0.001, *n* = 6) and by 25.03% (*p* < 0.001, *n* = 6), respectively, compared to the Sco group. In healthy rats, lipid peroxidation (LPO) levels were reduced in the cortex after *C. vulgare* treatment (by 54.08%, *p* < 0.001, *n* = 6) and in the hippocampus after *S. scardica* + *C. vulgare* combined treatment (by 35.82% (*p* < 0.001, *n* = 6). These results suggest that *C. vulgare* possessed the strongest antioxidant potential, visible in both healthy rats and rats with Sco-induced oxidative stress, correlating with the highest in vitro antioxidant activity of the extract.

#### 2.4.2. Effects on GSH Level

The effect of the treatment with plant extracts on glutathione (GSH) levels was assessed in the brain of healthy and Sco-treated rats (Figure 4A,B). Our results indicated that Sco application did not significantly change GSH content in the rat brains. In the cortex, the combination of *S. scardica* + *C. vulgare* treatment increased GSH content in both healthy (by 39.18%, *p* < 0.05, *n* = 6) and Sco-treated rats (by 57.12%, *p* < 0.001, *n* = 6) compared to the control and Sco-treated groups, respectively. In the hippocampus, GSH levels were evaluated after 21 days of *C. vulgare* administration. The increase was by 52.41% (*p* < 0.01, *n* = 6) in the healthy rats and by 37.67% (*p* < 0.05, *n* = 6) in the rats with dementia compared to the control and Sco-treated groups.

### 2.5. Effect of Plant Extracts on AChE Activity in the Frontal Cortex and Hippocampus of Healthy and Scopolamine-Treated Rats

The effect of plant extract administration on AChE activity in the frontal cortex and hippocampus of healthy and Sco-treated rats was investigated. As depicted in Figure 5A,B, Sco (2 mg/kg, i.p.) significantly elevated AChE activity in the hippocampus (by 35.28%, *p* < 0.05, *n* = 6) compared with the control, and did not significantly affect it in the cortex. The *C. vulgare* extract treatment of the rats with dementia preserved AChE activity in the hippocampus at the control level. However, *S. scardica* extract and the combination of *S. scardica* + *C. vulgare* did not exhibit AChE inhibitory activity in the cortex nor in the hippocampus.

### 2.6. Effect of Plant Extracts on the Content of Biogenic Amines in the Frontal Cortex and Hippocampus of Healthy and Scopolamine-Treated Rats

As shown in Figure 6, Sco administration for 11 days declined monoamine brain content in the experimental animals. In the cortex, there was a reduction of 62.26% (*p* < 0.001, *n* = 6) for NA and 24.85% (*p* < 0.05, *n* = 6) for Sero compared to the control group (Figure 6C–F). Regarding dopamine (DA), the effect was not statistically significant (Figure 6A,B). In the hippocampus, DA and noradrenalin (NA) were reduced by 21.12% (*p* < 0.05, *n* = 6) and 31.12% (*p* < 0.05, *n* = 6), respectively, compared with the control (Figure 6A–D). Sero was not significantly affected (Figure 6E,F).

After 21 days of plant extract treatment of the animals exhibiting dementia, a significant reversal of Sco-induced reduction of biogenic amines in the brain was observed. In the cortex of the animals with dementia, the combination of *S. scardica* + *C. vulgare* preserved the control levels of DA, NA and Sero. *S. scardica* treatment increased NA content by 135.23% (*p* < 0.01, *n* = 6) compared to the Sco-treated animals. In the hippocampus, the combination of *S. scardica* + *C. vulgare* preserved the control levels of NA. In healthy rats, the combination of *S. scardica* + *C. vulgare* increased DA levels in the frontal cortex by 40.25% (*p* < 0.05, *n* = 6) compared to the control (Figure 6A).

### 2.7. Effect of Plant Extracts on the Expression Levels of BDNF and pCREB in the Frontal Cortex and Hippocampus of Healthy and Scopolamine-Treated Rats

The expression of memory-related proteins BDNF and pCREB was quantified by ELISA methods. An analysis of the results showed that Sco administration led to a significant decrease in BDNF and pCREB expression levels in the frontal cortex and hippocampus (Figure 7A–D). In the cortex, the reduction was noteworthy, with a decrease of 28.44% (*p* < 0.001, *n* = 6) for BDNF and 45.24% (*p* < 0.001, *n* = 6) for pCREB when compared to saline-treated rats (Figure 7A,C). In the hippocampus, concentrations of BDNF and pCREB were reduced by 42.37% (*p* < 0.001, *n* = 6) and by 35.13% (*p* < 0.001, *n* = 6), respectively, compared to the control group (Figure 7B,D).

The administration of plant extracts to animals exhibiting dementia reversed Sco-induced changes in brain BDNF and pCREB concentrations. In the cortex, the BDNF concentration was elevated by 18.37% (*p* < 0.05, *n* = 6) in the *S. scardica*-treated group. pCREB expression was also increased by *S. scardica*, *C. vulgare* and the combination of *S. scardica* + *C. vulgare* by 50.18% (*p* < 0.01, *n* = 6), 76.77% (*p* < 0.001, *n* =6) and 89.79% (*p* < 0.001, *n* = 6), respectively, compared to the Sco group. In the hippocampus, BDNF expression increased 40.90% (*p* < 0.05, *n* = 6) with *S. scardica* and 40% (*p* < 0.05, *n* = 6) with the combination of *S. scardica* + *C. vulgare*. pCREB concentration rose 25.59% (*p* < 0.05, *n* = 6) with *C. vulgare* after 21 days of treatment. In healthy animals, ELISA analysis demonstrated that plant extract treatment did not induce significant change in p-CREB/BDNF signaling in the frontal cortex and hippocampus.

## 3. Discussion

In this study, we conducted a comparative analysis of the effects of *S. scardica*, *C. vulgare* and their combination on learning and memory performance in both healthy rats and rats with Sco-induced memory impairment. To elucidate the mechanisms underlying the impact of these plant extracts on memory processes, we evaluated their anti-AChE and antioxidant properties, along with their effect on biogenic amine levels in the rat cortex and hippocampus. Additionally, we investigated the effects of these plant extracts on the expression of two brain peptides, BDNF and pCREB, which are closely associated with memory processes.

Scopolamine, a muscarinic Ach receptor antagonist, is widely accepted as a pharmacological tool for inducing an experimental model of Alzheimer-type dementia [64]. It induces learning and memory impairments in animals and humans by elevating AChE activity, increasing oxidative stress, decreasing biogenic amine levels, and suppressing the expression of genes related to memory processes, such as *BDNF* and *CREB*, in the hippocampus and cortex [53,65,66,67,68].

To assess the impact of the selected plant extracts and their combination on learning and memory processes in healthy and demented animals, we conducted two behavioral tests: the T-maze and the NOR tests. Each behavioral experiment evaluates a different aspect of memory.

The T-maze test was employed to assess the spatial working memory of the animals, a function closely dependent upon hippocampal activity [69,70,71]. Spatial memory impairment constitutes a primary characteristic of the cognitive decline observed in AD patients [72]. In the second test, NOR, we examined deficits in the animal recognition memory. NOR is based on the innate tendency of animals to prefer exploring new objects over familiar ones and does not involve spatial reference memory [73], implying the presence of encoded memory for familiar objects.

Our results demonstrate that Sco significantly suppressed cognitive function, consistent with our previous reports [53,67,68,74]. The cognitive decline was evident through a significantly decreased number of correct choices in the T-maze test and reduced time spent with the new object in the NOR test. As anticipated, treatment with plant extracts ameliorated Sco-induced cognitive impairment. The most pronounced preservation of spatial working memory was observed in the group of animals with dementia treated with the extract of *S. scardica*, which resulted in a significant increase in the number of correct choices compared to the Sco-treated animals.

According to the results of the NOR test, the treatment of animals with dementia with selected plant extracts increased the time spent with the new object, indicating a memory-preserving effect. The combination of *S. scardica* + *C. vulgare* exhibited the best protective recognition memory effect. In healthy rats, treatment with plant extract did not induce significant changes in spatial working and recognition memory in the animals.

Oxidative stress is an additional factor in the etiology of AD and a contributing cause of memory impairment [75,76,77]. Although the origin of increased ROS production and the exact mechanisms underlying the disruption of redox balance in the disease progression remain elusive, the interrelationship between increased oxidative stress, mitochondrial dysfunction, energy failure and aggregation of Aβ and tau pathology in brains afflicted with AD is evident [78,79,80,81]. Furthermore, the increased generation of ROS is implicated in triggering cognitive disturbance in AD [82].

In this study, Sco administration increased the brain MDA levels but did not significantly alter GSH content in the cortex and hippocampus of the rats. MDA is a highly reactive and cytotoxic end product of tissue lipid peroxidation, serving as an indicator of escalating oxidative stress. However, treatment with plant extracts effectively restored the oxidative stress parameters altered by Sco. Our results show that *C. vulgare* treatment exhibited the most powerful antioxidant capacity. This plant extract significantly reduced LPO levels in the cortex and hippocampus and increased GSH content in the hippocampus of healthy rats and rats with dementia. In addition, the combination of *S. scardica* + *C. vulgare* also exhibited an antioxidant effect confined within the frontal cortex. These treatments successfully decreased MDA, the end product of lipid peroxidation, and increased GSH in healthy animals and animals with dementia.

The inhibitory action on brain oxidative damage and the beneficial memory effects of the plant extracts are most likely connected with the presence of phenols and flavonoids. Phenols (ArOH), a diverse group of metabolites originating from the secondary metabolism of plants, possess the outstanding property of suppressing or delaying the spontaneous autoxidation of organic molecules [83,84]. Polyphenols, particularly flavonoids, can traverse the blood–brain barrier, rendering them suitable as anti-AD natural agents [85]. The reported anti-amyloidogenic features, AChE inhibitory activity and cognitive decline-preventing effects of polyphenols and flavonoids [85,86,87] underscore their potential therapeutic relevance.

To elucidate the effect of two plant extracts (*S. scardica* and *C. vulgare*) and their combination on the acetylcholinergic neurotransmitter system in healthy rats and rats with experimental dementia, we assessed the changes in AChE activity in the cortex and hippocampus of the rodents. According to the cholinergic hypothesis of AD, the cognitive deficit, which is the primary characteristic of the disease, is attributed to the selective loss of cholinergic neurons in the nucleus basalis magnocellularis (NBM) and the resulting decrease in synaptic levels of ACh [88,89]. One of the primary strategies in the development of antidementia drugs is to enhance ACh levels in the synaptic cleft, typically achieved by inhibiting AChE to prevent the synaptic degradation of ACh. AChE inhibitors remain the main therapeutic approach for treating AD.

Scopolamine, acting as a muscarinic ACh receptor antagonist, significantly increases AChE activity in the cortex and hippocampus, contributing to its dementia-inducing effects [67,68,74]. However, according to our findings, *C. vulgare* treatment inhibited AChE activity in the hippocampus. These results confirm our previous findings that AChE inhibitory activity is not a contributing factor to the memory-enhancing effects of *S. scardica* [53]. In healthy animals, treatment with selected plant extracts did not significantly change the activity of AChE in the cortex and hippocampus. However, pathological brain changes in AD not only affect cholinergic neurotransmission but also damage DA, NA, Sero and glutamatergic systems [20,90,91,92].

To clarify whether *C. vulgare*, *S. scardica* and their combination affect the levels of biogenic amines in the brain, we measured the levels of DA, NA and Sero in the cortex and hippocampus of healthy rats and rats with Sco-induced dementia. Our results indicated that Sco administration resulted in a reduction of the content of biogenic amines in the cortex and hippocampus of the animals compared to the control group [67,74]. DA was most affected in the hippocampus, whereas NA and Sero were impacted in the cortex. However, some of the plant extracts exhibited a preserving effect on the brain monoaminergic system. The combination of *S. scradica* + *C. vulgare* sustained the control levels of all measured biogenic amines in the cortex and of NA in the hippocampus of rats with dementia. *S. scardica* treatment positively impacted NA content in the cortex of Sco-treated animals. In healthy animals, DA neurotransmission was increased by the combination of *S. scradica* + *C. vulgare* in the cortex.

Several clinical and experimental findings suggest that noradrenergic projections from the locus coeruleus regulate neurovascular coupling, and the degeneration of locus coeruleus neurons in AD subjects diminishes the ability to couple blood volume to oxygen demand [93]. Considering our results, which demonstrate the noradrenaline-preserving effect of *S. scardica*, we may hypothesize that the memory-enhancing effect of the extract is partly attributable to noradrenaline preservation and the modulation of cerebral blood flow. This aligns with the assumption by Wightman et al. [94] that the acute cognitive effects of *S. scardica* in patients are probably connected with the modulation of cerebral blood flow, with ferulic and chlorogenic acid identified as potentially active substances in the extract [94].

To further elucidate the memory-enhancing mechanisms of selected plant extracts, the expression levels of BDNF and pCREB were investigated by the ELISA method in the hippocampus and frontal cortex. p-CREB/BDNF signaling cascades are known to be involved in neuronal cell survival and long-term memory formation and storage [95]. BDNF, a growth factor in the hippocampal and cortical regions, is essential for neuronal cell protection, synaptic transmission, learning and memory [16,96]. BDNF expression is regulated by CREB [97]. In addition, a previous study indicated that Sco downregulated hippocampal and cortical p-CREB/BDNF protein expression, impairing memory in mice and rats [67,98]. Similarly, our observation revealed that Sco successfully decreased BDFN and phospho-CREB expression in the hippocampus and frontal cortex. However, the results from our ELISA analysis demonstrated that treatment with plant extracts mitigated the Sco-induced downregulation of BDNF and phospho-CREB proteins. *S. scardica* showed the best restored effect for BDNF, the combination of *S. scardica* and *C. vulgare* for pCREB in the cortex, and *C. vulgare* for pCREB in the hippocampus.

We therefore consider that the selected plant extracts may ameliorate learning and memory impairments induced by Sco by preserving the p-CREB/BDNF signaling pathway in the hippocampus and frontal cortex of rats.

In conclusion, our research highlights the notable effectiveness of the combination of *S. scardica* + *C. vulgare*, manifesting the best recognition memory preservation, strong antioxidant activity and a superior monoaminergic preserving effect in the cortex. *S. scardica* exhibited remarkable spatial working memory preservation, while *C. vulgare* demonstrated recognition memory preservation comparable to *S. scardica*, coupled with the strongest antioxidant effect and AChE inhibitory activity. Importantly, all plant extracts effectively counteracted the Sco-induced downregulation of p-CREB/BDNF signaling.

Our analysis revealed that *S. scardica* and *C. vulgare* extracts are particularly rich in phenolic compounds. The largest portion of polyphenols in *S. scardica* consists of phe-nylpropanoids derivatives, including verbascoside and forsythoside A, among many others [99]. Forsythoside A, notable for its neuroprotective properties, has been shown to mitigate Alzheimer’s-like pathology by inhibiting ferroptosis-mediated neuroinflammation [100]. Furthermore, a clinical study has demonstrated the bioavailability of polyphenols in *S. scardica*; around five percent of the polyphenols ingested with a cup of tea were found as metabolites in urine samples [101]. The metabolite profile of *C. vulgare* reveals that caffeic acid, chlorogenic acid and catechin are the most abundant secondary metabolites in the plant. It appears that the caffeoyl moiety in the structure of phenolic acids is essential for their anti-inflammatory activity [62].

To summarize, our research intricately linked the impact of plant extracts on learning and memory performance in conditions of experimental dementia to their diverse effects on the monoaminergic neurotransmitter system, the antioxidant defense system of cells and p-CREB/BDNF signaling in the hippocampus and frontal cortex. Specifically, in healthy animals, *C. vulgare* and the combined treatment with *S. scardica* + *C. vulgare* demonstrated a potential for synergistically stimulating DA neurotransmission in the cortex. These findings underscore the necessity for further investigations to explore the specific molecular mechanisms behind these and to establish optimal dosage regimens for practical applications in dementia treatment. Additionally, investigating the potential synergistic effects of combining specific plant extracts could offer valuable insights for developing targeted therapeutic interventions for cognitive disorders.

However, it is important to recognize a key limitation in our methodology: reliance on a single dose, which is crucial in understanding the efficacy and pharmacological profile of the studied extracts. Different doses can elicit varied pharmacological responses, and understanding the dose–response relationship is essential for identifying the most effective and safe therapeutic window. Therefore, despite the fact that our findings contribute valuable insights into the effects of these plant extracts on learning and memory, the single dose approach limits the scope of pharmacological understanding. Future studies should explore a range of dosages, which would not only reinforce our findings but also aid in comprehensively assessing the pharmacological potential and clinical applicability of these extracts in treating dementia.

## 4. Materials and Methods

### 4.1. Plant Material

*Sideritis scardica* and *Clinopodium vulgare* plants were obtained by in vitro methods and subsequently cultivated on the experimental field in Elin Pelin (located near Sofia city, at altitude of 560 m) [63,102]. The aerial parts of the plants were collected during the summer months (June–July) and dried in the shade at room temperature (19–21 °C).

### 4.2. Preparation of Freeze-Dried Extracts

Approximately 50 g of dried aerial parts from *S. scardica* and *C. vulgare* plants were finely ground using a laboratory grinder. Five grams of the resulting powder were then added to 200 mL water at 90 °C and extracted for 15 min. The suspension was further centrifuged at 6000× *g*, and the supernatants were collected and freeze-dried for 96 h using an Alpha 1–4 LD plus laboratory freeze drier (Martin Christ Gefriertrocknungsanlagen GmbH, Osterode am Harz, Germany).

### 4.3. Total Polyphenol and Total Flavonoid Content

The total polyphenol content of the freeze-dried extracts was determined following the method of Singleton and Rossi [103] and expressed in mg gallic acid equivalents (GAE) per 100 g of dry weight (DW) ± SD. Total flavonoid content was assessed according to Chang et al. [104] and expressed as mg rutin equivalents (RE) per 100 g of DW ± SD.

### 4.4. Antioxidant Activity Assays

The oxygen radical absorbance capacity (ORAC) and hydroxyl radical averting capacity (HORAC) of the freeze-dried extracts were measured following the protocols outlined by Ou et al. [105,106] with some modifications as detailed by Denev et al. [107]. Both assays were conducted using a FLUOstar OPTIMA microplate reader (BMG Labtech, Ortenberg, Germany) with excitation at λ = 485 nm and emission at λ = 520 nm. The results were quantified and expressed in micromole trolox equivalents (μmol TE) and micromole gallic acid equivalents (μmol GAE) per gram of DW ± SD for ORAC and HORAC, respectively.

### 4.5. Animals

Male Wistar rats, weighing 180–200 g at the beginning of the experiment, were obtained from Erboj, Slivniza, Sofia. The rats were housed six per cage in a temperature and light-controlled environment (25 ± 2 °C, a 12-h light-dark cycle) with ad libitum access to food and water. A habituation period of 5 days preceded the initiation of the experiment. The experimental protocol adhered to the requirements of the European Communities Council Directive (86/609/EEC). Substantial efforts were undertaken to minimize animal suffering and reduce the overall number of animals used in the experiment. Behavioral experiments were performed during the morning hours (8–12 h) with the light level in the experimental room set at 200 lux.

### 4.6. Experimental Design

The animals were randomly divided into 8 experimental groups (*n* = 18 per group): (1) control group; (2) Sco group; (3) *S. scardica* group; (4) *C. vulgare* group; (5) *S. scardica* + *C. vulgare* group; (6) Sco + *S. scardica* group; (7) Sco + *C. vulgare* group; (8) Sco + *S. scardica* + *C. vulgare* group.


The control, *S. scardica*, *C. vulgare* and *S. scardica* + *C. vulgare* groups received intraperitoneal (i.p.) saline injection (0.9% NaCl, 0.5 mL/100 g b.w.).The Sco, Sco + *S. scardica*, Sco + *C. vulgare* and Sco + *S. scardica* + *C. vulgare* groups were injected with Sco hydrobromide at a dose of 2 mg/kg, i.p.The control and Sco groups received distilled water (dH_2_0) orally (0.5 mL/100 g b.w.).The *S. scardica*, *C. vulgare*, *S. scardica* + *C. vulgare*, Sco + *S. scardica*, Sco + *C. vulgare* and Sco + *S. scardica* + *C. vulgare* groups received plant extracts orally (0.5 mL/100 g b.w.).


For the development of the experimental model of Alzheimer-type dementia, the rats were injected with Sco hydrobromide (2 mg/kg, i.p.) for 11 consecutive days [67,74,108,109,110,111]. Scopolamine was dissolved in dH_2_0 ex tempore before each administration.

Plant extracts were administered at a dose of 200 mg/kg for *S. scardica* [112], 100 mg/kg for *C. vulgare* [113] and in combination (1:1). The treatment regimen spanned 21 consecutive days, starting 10 days before and continuing for 11 days simultaneously with Sco administration. The extracts were given to the rats one hour before the Sco injection (Figure 8).

### 4.7. Behavioral Assessment

#### 4.7.1. T-Maze Test

Spatial memory was assessed through the T-maze test by evaluating spontaneous alternation behavior. The experimental animals were tested in a rewarded alternation paradigm in an enclosed maze, following a previously described method [114]. Each animal underwent training and test sessions. The T-maze apparatus (Stoelting Co., Wood Dale, IL, USA) consisted of a start arm (50 × 10 cm) and two short arms (40 × 20 cm) arranged in a T-shape. Training session: Before the training session, the experimental animals underwent two days of handling by the experimenter and two days of habituation to the T-maze apparatus with the purpose of exploring the maze. During the training session, each animal was initially placed in the enclosure of the starting arm and released after a 10-s interval. The first trial was a forced trial with a reward pellet positioned at the end of the open right arm, while the opposite arm was obstructed by a guillotine door. In the following trial, both arms were opened, but only the opposite arm to that baited during the forced trial was rewarded. Each set consisted of 10 trials with a five-minute gap between them. A correct choice was recorded when the animals visited the baited arm. The training continued until a set achieved a 90% positive response. The training session spanned 5 days before all treatments. Test session: The T-maze test was carried out on the 22nd day after the first plant extract treatment. The animals performed 10 trials, with a five-minute interval between each trial and with a forced first attempt (on the right). The data are presented as a percentage of the number of correct choices divided by the total number of trials minus 1, multiplied by 100.

#### 4.7.2. Novel Object Recognition (NOR) Test

The NOR test is a well-established behavioral assessment used to evaluate hippocampus-related recognition memory based on the natural inclination of animals to allocate more time investigating a novel object compared to a familiar one [115]. The test procedure includes two main phases: habituation and the test session. The habituation phase took place one day before the real test. The rats were placed in a box (60 × 60 × 60 cm) with two identical objects and allowed to explore them for a duration of 3 min. The test session was initiated 24 h after habituation and consisted of two steps. In the first step, the rats were given 4 min to explore two identical objects within the recognition box. The second step commenced 1 h after the first one. The researcher reintroduced the rats to the recognition box, but with two different objects: one was familiar, and the other novel. Each animal was allotted 3 min to explore these two objects freely.

To assess recognition memory, the discrimination index is calculated using the following formula: discrimination index (DI) = (time spent on the novel object/total time spent exploring both objects) × 100%. The exploration of an object was defined as the animal placing its nose within 2 cm of the zone where the object was located.

### 4.8. Brain Dissection Technique

After completing the behavioral assessments, the rats were decapitated, and their brains were carefully removed from the skull for analysis. Initially, the skin covering the skull was incised and removed to expose the dorsal skull plates. These plates were then split along the midline, twisted and turned across the lateral border to facilitate brain extraction. For hippocampus isolation, two short microspatulas were used. The tip of one spatula was secured just above the cerebellum near the junction with the cortex. Simultaneously, the tip of the other spatula gently peeled the cortical hemisphere laterally, revealing the hippocampus, which was then carefully “scooped” laterally. Subsequently, the frontal cortex was dissected. During the entire procedure, the tissue was periodically rinsed with fresh ice-cold saline (0.9% NaCl) using a pipette.

### 4.9. Analytical Assessment

For the biochemical analyses, the animals were euthanized one hour after the last behavioral test by decapitation under mild CO_2_ inhalation. The cortex and hippocampus were quickly separated from the brain, cleaned with ice-cold saline and stored at –20 °C.

#### 4.9.1. Oxidative Stress Parameters Determination

The levels of LPO and GSH were measured in post-nuclear fractions, which were prepared by centrifuging 0.15 M KCl homogenates from the frontal cortex and hippocampus at 3000× *g* for 10 min at 4 °C. The activity of antioxidant enzymes was measured in a post-mitochondrial fraction obtained through additional centrifugation of a portion of the post-nuclear fraction at 15,000× *g* for 20 min at 4 °C.

##### Lipid Peroxidation and Total Glutathione Determination

The concentrations of MDA and total GSH in the frontal cortex and hippocampus of the rats were determined using ELISA methods according to the manufacturer instructions. The following commercial kits were used: Rat TBARS ELISA kit (Cat. no. MAK085); Rat GSH ELISA kit (Cat. no. CS0260), purchased from Sigma-Aldrich Co. LLC, St. Louis, MO, USA. A thiobarbituric acid (TBARS) activity assay was performed to assess the brain MDA level. The TBARS and GSH concentrations were measured spectrophotometrically at 532 nm and 412 nm absorption, respectively. The MDA concentration was expressed as nmoles MDA/mg protein, with a molar extinction coefficient of 1.56 × 105 M^−1^ cm^−1^. The GSH concentration was calculated from the reference standard and expressed as ng/mg protein.

#### 4.9.2. Protein Content Determination

The protein contents in the frontal cortex and hippocampus of the rats were determined by the method of Lowry [116]. Protein concentrations were measured spectrophotometrically at the 700 nm absorption peak and were determined using a calibration curve obtained with bovine serum albumin (Pentex, Miles Laboratories, Elkhart, IN, USA).

#### 4.9.3. AChE Activity Assay Determination

The AChE activity in the frontal cortex and hippocampus of the rats was evaluated using Ellman’s method [117]. Briefly, a 10% tissue homogenate was centrifuged at 4500 rpm for 10 min. Subsequently, 100 µL of the supernatant was incubated with the Ellman reagent, consisting of 0.01 M DTNB, 0.1 M phosphate buffer and 0.075 M freshly prepared acetylthiocholine iodide. The kinetics of the reaction were monitored for 3 min at 405 nm using a semi-auto chemistry analyzer.

#### 4.9.4. Monoamines Content Determination

The concentration of biogenic amines, noradrenalin (NA), dopamine (DA) and serotonin (Sero) in brain tissue were measured using the method established by Jacobowitz and Richardson [118]. The extraction of NA and DA was carried out using a phosphate buffer. The fluorescence reaction necessitated the use of ethylenediaminetetraacetic acid (EDTA), iodide solution, alkaline sulfite and 5N CH_3_COOH. Fluorescence intensity was measured at λ = 385/485 nm for NA and at λ = 320/385 nm for DA. For Sero, the extraction was performed using 0.1 N HCl. The fluorescence reaction involved o-phthaldehyde and fluorescence intensity was measured at λ = 360/470 nm. Monoamine fluorescence levels were then calculated based on the fluorescence of the standard solution and expressed as ng/g of fresh tissue.

#### 4.9.5. Determination of BDNF and pCREB Concentrations

The concentrations of Brain-Derived Neurotrophic Factor (BDNF) and phosphorylated cAMP in the frontal cortex and hippocampus of rats were quantified by ELISA, following the manufacturer instructions. The following kits were used: Rat BDNF ELISA Kit (cat. no. E-EL-R1235); Rat Phospho cAMP response element binding protein (P-CREB) ELISA Kit (cat. no. SL1344Ra), purchased from Elabscience (Houston, TX, USA). The concentrations were measured using a microplate reader at 450 nm. The results were reported in picograms per milliliter (pg/mL).

### 4.10. Statistical Analysis

The data are presented as mean ± standard error of the mean (SEM) and were analyzed with Prism 8.0 software (GraphPad Software, Inc., San Diego, CA, USA). Statistical analyses were conducted through one-way analysis of variance (ANOVA) followed by Tukey’s post hoc comparison test. Differences were considered significant at *p* < 0.05. The results of biological activity tests were calculated using Microsoft Excel (Microsoft Corporation, Redmond, WA, USA) and expressed as means ± standard deviations (SD). All experiments were repeated 3 times.

## 5. Conclusions

Our research highlights the combination of *S. scardica* + *C. vulgare*, noting that it showed the best recognition memory-preserving effect, strong antioxidant activity and the best monoaminergic activity-preserving effect in the cortex. *S. scardica* exhibited the most notable spatial working memory-preserving effect. *C. vulgare* showed a recognition memory-preserving effect commensurate with that of *S. scardica*, along with the strongest antioxidant activity and AChE inhibitory activity. Importantly, all plant extracts mitigated the scopolamine-induced downregulation of p-CREB/BDNF signaling. In healthy animals, *C. vulgare* and the combination of *S. scardica* + *C. vulgare* treatment stood out for their antioxidant activity and the potential to synergistically stimulate DA neurotransmission in the cortex.

We can conclude that the effects of plant extracts on learning and memory performance in the experimental dementia condition are partly connected with their multiple effects on the monoaminergic neurotransmitter system, the antioxidant defense system of the cells and p-CREB/BDNF signaling in the hippocampus and frontal cortex. Additionally, investigating potential synergistic effects when combining specific plant extracts could offer valuable insights for the development of targeted therapeutic interventions for cognitive disorders.

## Figures and Tables

**Figure 1 ijms-25-01840-f001:**
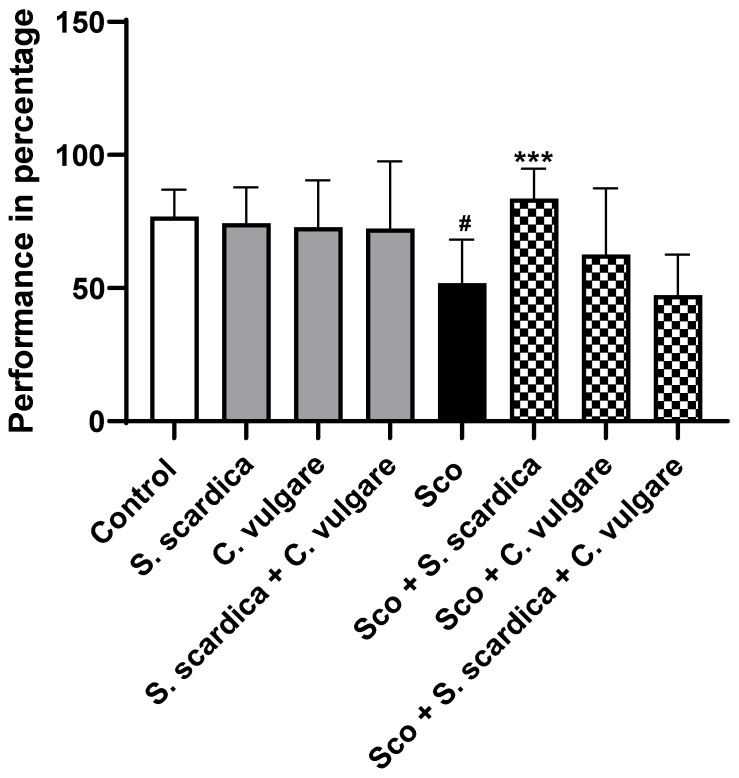
Effect of *S. scardica*, *C. vulgare* extracts and their combination on rewarded spontaneous alternation behavior of healthy rats and rats with scopolamine-induced memory impairment in the T-maze test. Results are presented as mean values ± SEM (*n* = 10 animals per group). Statistical analysis involved one-way analysis of variance (ANOVA), followed by Tukey’s multiple comparison test. Significance vs. saline-treated group: ^#^ *p* < 0.05; significance vs. Sco-treated group *** *p* < 0.001.

**Figure 2 ijms-25-01840-f002:**
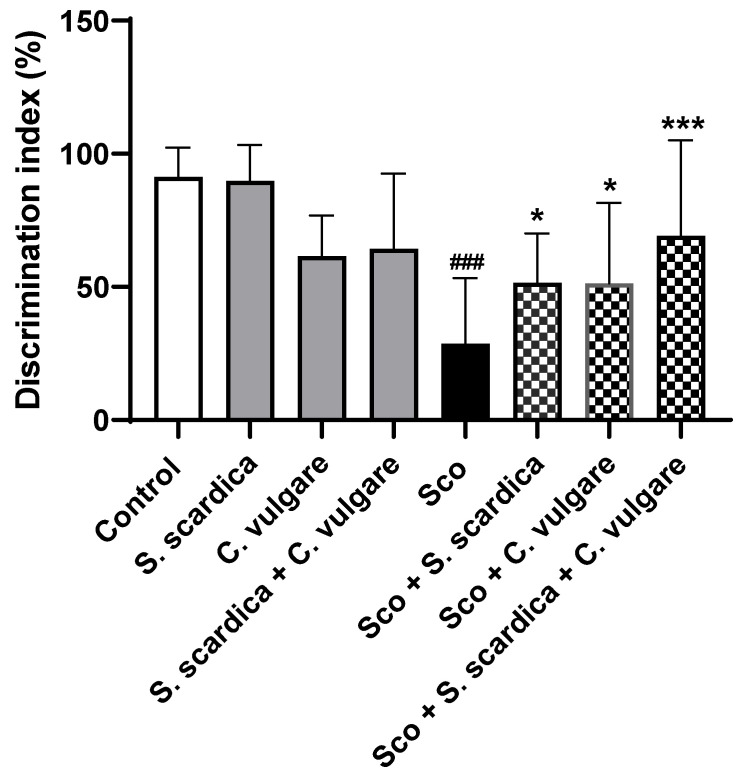
Effect of *S. scardica*, *C. vulgare* extracts and their combination on recognition memory of healthy rats and rats with scopolamine-induced memory impairment in the NOR test. The plant extracts were administered for 21 days. Results are presented as mean values ± SEM (*n* = 8 animals per group). Statistical analysis involved one-way analysis of variance (ANOVA), followed by Tukey’s multiple comparison test. Significance vs. saline-treated group: ^###^ *p* < 0.001; significance vs. Sco-treated group * *p* < 0.05, *** *p* < 0.001.

**Figure 3 ijms-25-01840-f003:**
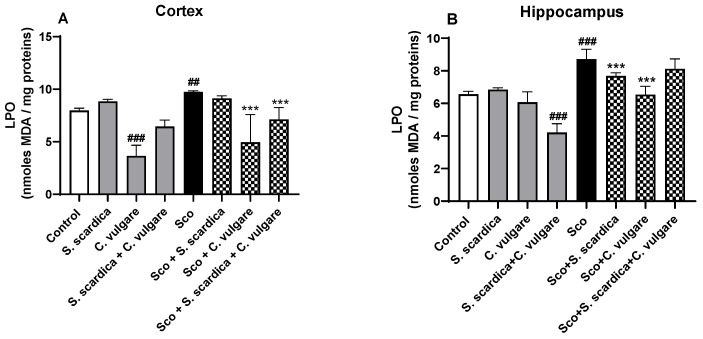
Effect of *S. scardica*, *C. vulgare* extracts and their combination on LPO levels in the cortex (**A**) and hippocampus (**B**) of healthy rats and rats with scopolamine-induced memory impairment. Results are presented as mean values ± SEM (*n* = 6 animals per group). Statistical analysis involved one-way analysis of variance (ANOVA), followed by Tukey’s multiple comparison test. Significance vs. saline-treated group: ^##^ *p* < 0.01, ^###^ *p* < 0.001; significance vs. Sco-treated group *** *p* < 0.001.

**Figure 4 ijms-25-01840-f004:**
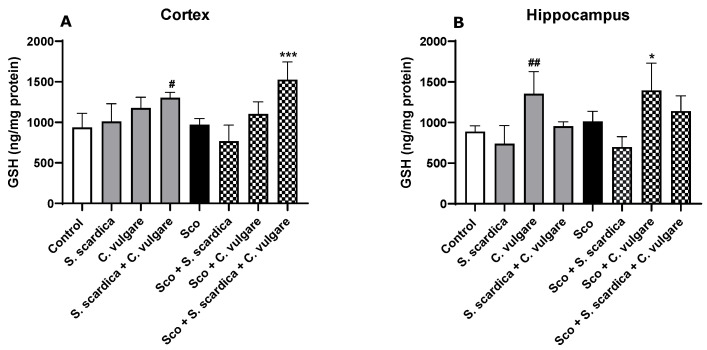
Effect of *S. scardica*, *C. vulgare* extracts and their combination on GSH levels in the cortex (**A**) and hippocampus (**B**) of healthy rats and rats with scopolamine-induced memory impairment. Results are presented as mean values ± SEM (*n* = 6 animals per group). Statistical analysis involved one-way analysis of variance (ANOVA), followed by Tukey’s multiple comparison test. Significance vs. saline-treated group: ^#^ *p* < 0.05, ^##^ *p* < 0.01; significance vs. Sco-treated group * *p* < 0.05, *** *p* < 0.001.

**Figure 5 ijms-25-01840-f005:**
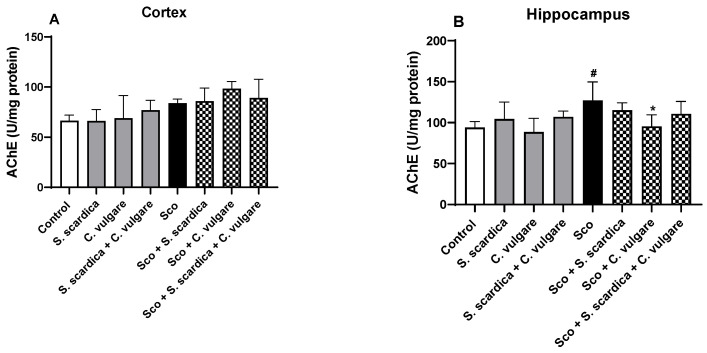
Effect of *S. scardica*, *C. vulgare* extracts and their combination on brain AChE activity of healthy rats and rats with scopolamine-induced memory impairment. The enzyme activity was assessed in cortex (**A**) and hippocampus (**B**) of the rats. Results are presented as mean values ± SEM (*n* = 6 animals per group). Statistical analysis involved one-way analysis of variance (ANOVA), followed by Tukey’s multiple comparison test. Significance vs. saline-treated group: # *p* < 0.05; significance vs. Sco-treated group * *p* < 0.05.

**Figure 6 ijms-25-01840-f006:**
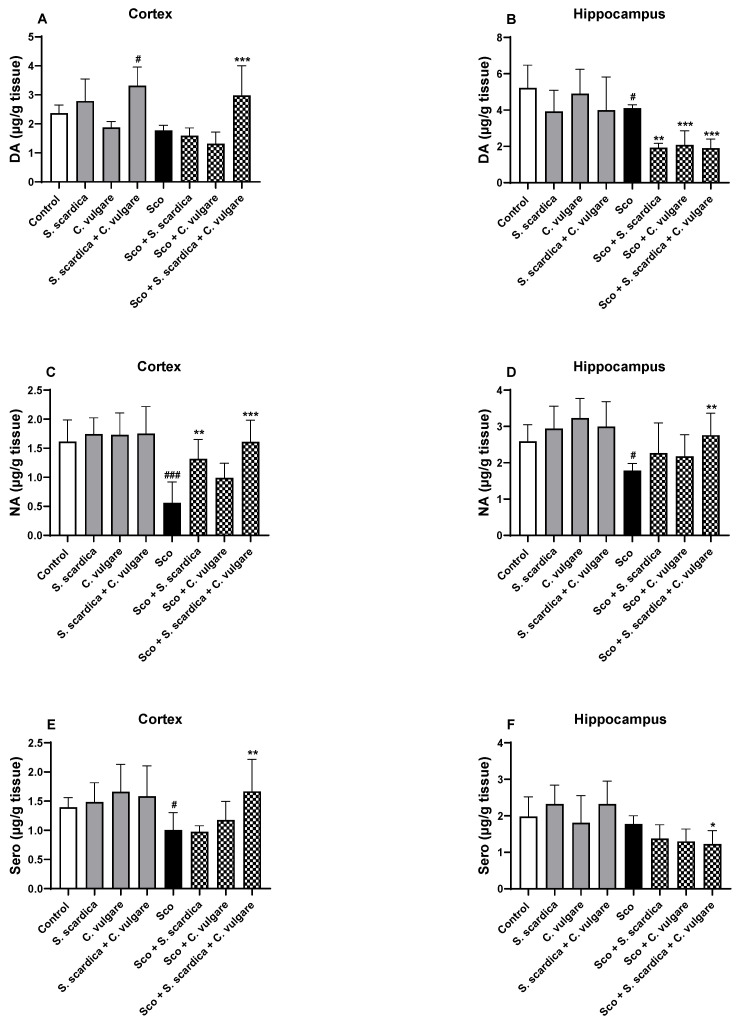
Effect of *S. scardica*, *C. vulgare* extracts and their combination on the content of brain biogenic amines in healthy rats and rats with scopolamine-induced memory impairment. The content of DA (**A**,**B**), NA (**C**,**D**) and Sero (**E**,**F**) content was assessed in the cortex and hippocampus of the rats. Results are presented as mean values ± SEM (*n* = 6 animals per group). Statistical analysis involved one-way analysis of variance (ANOVA), followed by Tukey’s multiple comparison test. Significance vs. saline-treated group: ^#^
*p* < 0.05, ^###^ *p* < 0.001 significance vs. Sco-treated group * *p* < 0.05, ** *p* < 0.01, *** *p* < 0.001.

**Figure 7 ijms-25-01840-f007:**
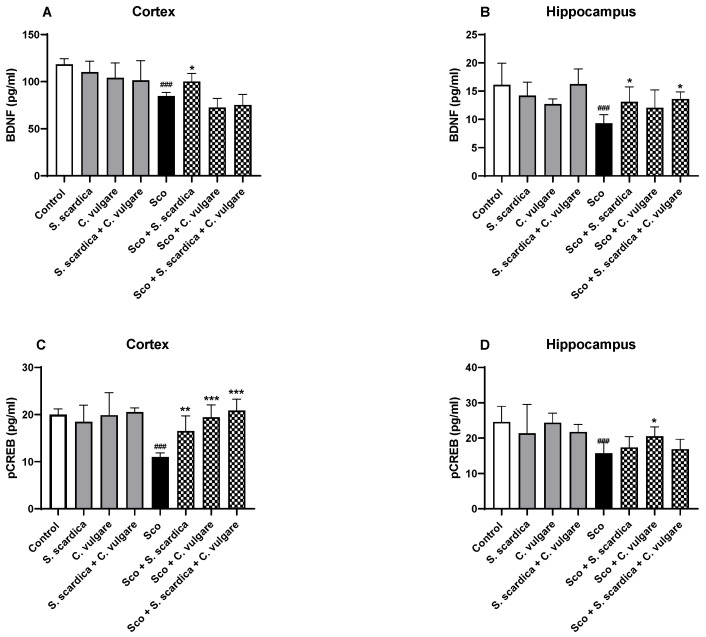
Effect of *S. scardica*, *C. vulgare* extracts and their combination on the brain concentrations of BDNF and pCREB in healthy rats and rats with scopolamine-induced memory impairment. The BDNF and pCREB concentrations were assessed in the cortex (**A**,**C**) and hippocampus (**B**,**D**) of the rats with enzyme-linked immunosorbent assay (ELISA). Results are presented as mean values ± SEM (*n* = 6 animals per group). Statistical analysis involved one-way analysis of variance (ANOVA), followed by Tukey’s multiple comparison test. Significance vs. saline-treated group: ^###^ *p* < 0.001 significance vs. Sco-treated group * *p* < 0.05, ** *p* < 0.01, *** *p* < 0.001.

**Figure 8 ijms-25-01840-f008:**
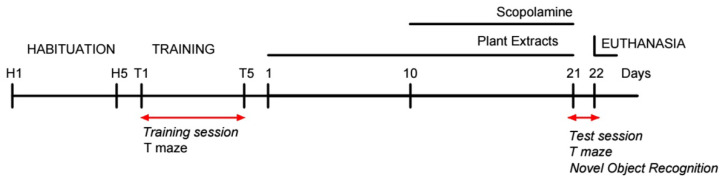
Experimental design.

**Table 1 ijms-25-01840-t001:** Polyphenol and flavonoid content and in vitro antioxidant activity of *S. scardica* and *C. vulgare* extracts.

Extract	Total Polyphenols (mg GAE/100 g DW)	Total Flavonoids (mg RE/100 g DW)	ORAC (µmol TE/g DW)	HORAC (µmol GAE/g DW)
*S. scardica* *	12,096 ± 1208 ^a^	1903 ± 229 ^c^	2595 ± 34 ^a^	718 ± 4 ^a^
*C. vulgare*	22,402 ± 812 ^c^	3689 ± 190 ^c^	6119 ± 195 ^c^	1538 ± 89 ^c^
Combination	15,064 ± 307 ^b^	2755 ± 102 ^b^	4698 ± 107 ^b^	1118 ± 95 ^b^

The data are presented as means of 3 samples ± standard deviation (SD). Different letters denote significant differences assessed by Fisher LSD test (*p* ≤ 0.05) after performing ANOVA multifactor analysis. * Results are derived from Tasheva et al. [63].

## Data Availability

All data are comprised in the manuscript.

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
