# Peer review of "Extracts of *Sideritis scardica* and *Clinopodium vulgare* Alleviate Cognitive Impairments in Scopolamine-Induced Rat Dementia"

_ijms, 2024, doi:10.3390/ijms25031840_

Round 1
Reviewer 1 Report
Comments and Suggestions for Authors
In this manuscript, the authors investigated the potential effect of C. vulgare on learning and memory in healthy and scopolamine (Sco)-induced memory-impaired male Wistar rats.
This is a potentially interesting study, however there are few concerns that should be addressed:
- The introduction is too long;
- Does exposure to plant extracts affects animal weight and food consumption?
- Given that 6 animals per group were used for each molecular experiment, how were the initial 18 animals per group divided?
- The AA should more precisely describe how they dissected the two brain regions.
Comments on the Quality of English Languageit would be necessary to have more details on the number of animals used for the molecular experiments.
Author Response
Reviewer 1: Comments and Replies
We sincerely thank the Reviewer for his diligent work and are grateful for the high appreciation of our manuscript.
Review Report: In this manuscript, the authors investigated the potential effect of C. vulgare on learning and memory in healthy and scopolamine (Sco)-induced memory-impaired male Wistar rats.
This is a potentially interesting study, however there are few concerns that should be addressed:
Reviewer: The introduction is too long
Authors’ Reply: We appreciate the Reviewer feedback on the length of the Introduction. The content in this section has been carefully chosen to lay a solid foundation for the rest of the manuscript. We have ensured that the text adheres to the journal guidelines regarding length and depth. However, in response to the Reviewer comment, we have made efforts to slightly refine this section while preserving its essential elements (pages 2 and 3 in the revised version).
Reviewer: Does exposure to plant extracts affects animal weight and food consumption?
Authors’ Reply: Treating the animals with the plant extracts under investigation did not significantly impact either animal weight or food consumption. Therefore, this particular set of data was deemed non-influential to the manuscript and was consequently excluded.
Reviewer: Given that 6 animals per group were used for each molecular experiment, how were the initial 18 animals per group divided?
Authors’ Reply: We appreciate the Reviewer inquiry regarding the division of the initial 18 animals per group for each molecular experiment. To clarify, the allocation of the animals was as follows:
- Six animals were designated for assessing oxidative stress parameters;
- Another set of six animals was utilized for measuring the levels of BDNF, pCREB and AChE activity;
- The final group of six animals was employed to determine catecholamine levels in the respective tissue homogenates.
This distribution ensured a balanced and focused approach to examining the various molecular aspects under study.
Reviewer: The AA should more precisely describe how they dissected the two brain regions.
Authors’ Reply: In response to this suggestion, we have thoroughly detailed the methodology for dissecting the two brain regions, including a step-by-step description of the dissection process. The new information has been included in the Material and Methods section of the revised manuscript on page 16. We hope this ensures clarity and reproducibility of the technique.
Reviewer 2 Report
Comments and Suggestions for Authors
In some figures the SEM of some groups have overlaps with each other, but you considered them as significant. would you please explain about them?
Author Response
Reviewer 2: Comments and Replies
Thank you to the reviewer for thorough review of our manuscript. The positive feedback is deeply appreciated and highly encouraging.
Reviewer: In some figures the SEM of some groups have overlaps with each other, but you considered them as significant. would you please explain about them?
Authors’ Reply: Thank you for your question about the SEM overlaps in our figures.
In our study, we employed one-way ANOVA, followed by Tukey's post-hoc multiple comparison test to compare the mean of each group with every other group. This approach ensures that statistical significance is determined by considering not only the overlap of SEM but also the actual means and variances of the groups. Tukey's post-hoc test, in particular, is crucial in adjusting for multiple comparisons, and in reducing the risk of Type I errors (false positives). Thus, even with some SEM overlap, differences between group means can be statistically significant.
Data analysis was conducted using Prism 8.0 software, recognized in the scientific community for its robust statistical analysis and graphing capabilities. Its advanced algorithms enabled us to accurately assess the significance of differences between groups, taking into account the entire dataset. The graphical representations in our manuscript show SEM, but primarily serve as a visual summary. Our core conclusions regarding statistical significance are based on the comprehensive statistical analysis detailed above, rather than solely on these graphical elements.
We believe this explanation clarifies any concerns regarding SEM overlaps and the determination of statistical significance in our study.

Reviewer 3 Report
Comments and Suggestions for Authors
The study addresses the very important issue of finding new natural remedies for dementia. The study is well done and methodologically quite adequate. However, there are some minor issues that need to be addressed. First of all, it is necessary to scientifically justify the choice of doses of S. scardica, C. vulgare and scopolamine used, possibly providing supporting literature, and the choice to evaluate the joint administration of the two herbs. What is the advantage of this? Could they interact? Secondly, a paragraph on the limitations of the study, which is now completely missing, needs to be added to the discussion. In particular, the use of a single dose reduces the pharmacological value of the results; it would have been useful to test several different doses for each substance in order to identify a dose-response effect.
Author Response
Reviewer 3: Comments and Replies
We are grateful for the Reviewer positive feedback and constructive comments on our study.
Reviewer: The study addresses the very important issue of finding new natural remedies for dementia. The study is well done and methodologically quite adequate. However, there are some minor issues that need to be addressed. First of all, it is necessary to scientifically justify the choice of doses of S. scardica, C. vulgare and scopolamine used, possibly providing supporting literature, and the choice to evaluate the joint administration of the two herbs. What is the advantage of this? Could they interact? Secondly, a paragraph on the limitations of the study, which is now completely missing, needs to be added to the discussion. In particular, the use of a single dose reduces the pharmacological value of the results; it would have been useful to test several different doses for each substance in order to identify a dose-response effect.
Authors’ reply: Addressing the first point, the dosages for S. scardica, C. vulgare and scopolamine were determined based on extensive literature review and prior empirical evidence. Scopolamine hydrobromide was administered to rats at 2 mg/kg, intraperitoneally, for 11 consecutive days, in line with references [67,74,109-112]. To ensure its effectiveness, scopolamine was freshly dissolved in distilled water before each administration.
Concerning the plant extracts, S. scardica was given at 200 mg/kg and C. vulgare at 100 mg/kg, following dosages recommended in existing literature (refs [113] and [114], respectively). These extracts were combined in a 1:1 ratio and administered for 21 consecutive days, starting 10 days before and continuing through the 11-day scopolamine treatment, one hour before each scopolamine injection. The rationale behind evaluating the joint administration of these herbs was to investigate any potential synergistic effects that might enhance therapeutic efficacy.
Regarding the second point, we recognize that using single doses may restrict the pharmacological interpretation of the results, and consider the need for further investigation into their interactive properties. Consequently, a brief section discussing the limitations of our study has now been added to the end of the Discussion part (page 13).
- Tancheva, L.; Lazarova, M.; Velkova, L.; Dolashki, A.; Uzunova, D.; Minchev, B.; Petkova-Kirova, P. Hassanova, Y.; Gavrilova, P.; Tasheva, K.; et al. Beneficial effects of snail Helix aspersa extract in an experimental model of Alzheimer's type dementia. J. Alzheimers Dis. 2022, 88(1), 155–175.
74. Staykov, H.; Lazarova, M.; Hassanova, Y.; Stefanova, M.; Tancheva, L.; Nikolov, R. Neuromodulatory mechanisms of a memory loss-preventive effect of alpha-lipoic acid in an experimental rat model of dementia. J. Mol. Neurosci. 2022, 72(5), 1018–1025.
109. Lee, B.; Sur, B.; Shim, I.; Lee, H.; Hahm, D. Phellodendron amurense and its major alkaloid compound, berberine ameliorates scopolamine-induced neuronal impairment and memory dysfunction in rats. Korean J. Physiol. Pharmacol. 2012, 16(2), 79–89.
110. Upadhyay, P.; Shukla, R.; Kavindra Tiwari, K.N.; Dubey, G.P.; Mishra, S.K. Neuroprotective effect of Reinwardtia indica against scopolamine induced memory-impairment in rat by attenuating oxidative stress. Metab. Brain Dis. 2020, 35(5), 709–725.
111. Shivakumar, S.; Ilango, K.; Agrawal, A.; Dubey, G.P. Efect of hippophae rhamnoides on cognitive enhancement via neurochemical modulation in scopolamine induced Sprague Dawely rats. Int. J. Pharm. Sci. Res. 2014, 6, 4153–4158
112. Tsvetanova, E.; Alexandrova, A.; Georgieva, A.; Tancheva, L.; Lazarova, M.; Dolashka, P.; Velkova, L.; Dolashki, A.; Atanasov, V.; Kalfn, R. Efect of mucus extract of Helix aspersa on scopolamineinduced cognitive impairment and oxidative stress in rat’s brain. Bulg. Chem. Commun. 2020, 52, 107–111
113. Jeremic, I.; Petricevic, S.; Tadic, V.; Petrovic, D.; Tosic, J.; Stanojevic, Z.; Petronijevic, M.; Vidicevic, S.; Trajkovic, V.; Isakovic, A. Effects of Sideritis scardica extract on glucose tolerance, triglyceride levels and markers of oxidative stress in ovariectomized rats. Planta Med. 2019, 85(6), 465–472.
114. Nasar-Eddin, G.; Simeonova, R.; Zheleva-Dimitrova, D.; Gevrenova, R.; Savov, I.; Bardarov, K.; Danchev, N. Beneficial effects of Clinopodium vulgare water extract on spontaneously hypertensive rats. Bul Chem Comm 2019, 51(A), 156–160.
Reviewer 4 Report
Comments and Suggestions for Authors
Maria Lazarova et al. study on Extracts of Sideritis scardica and Clinopodium vulgare Alleviate Cognitive Impairments in Scopolamine-Induced Rat Dementia is designed well and performed nicely, but still the manuscript needs some improvement and can be accepted after major revision. The comments and suggestions are given below:-
Comment 1: Is the concentration mentioned by the author in the article the optimal therapeutic concentration? How do you determine which concentration to use? Are there other concentrations that have better results?
Comment 2: The authors mentioned that Sco-induced downregulation of p-CREB/BDNF signaling demonstrated the protective effect of extracts. How can the neuroprotective effect be determined from the results presented that do not demonstrate upregulation of p-CREB/BDNF signaling after crude extract treatment? Relevant evidence should be provided that p-CREB/BDNF signalling is involved in neuroprotection.
Comment 3: This study found that C. vulgare S. scardica and their combination can improve learning and memory abilities, but there is no evidence of its potential mechanism of action. Some experiments are needed to prove its mechanism of action.
Comment 4:The main components of S. scardica, C.vulgare, and their combination are phenolic substances. Which type of bioactive components play the main role? The neuroprotective effect should be further verified in vitro using hippocampal neuron cell models (such as HT22 cells).
Author Response
Reviewer 4: Comments and Replies
Thank you very much for your helpful feedback, which has helped us improve our manuscript. We believe the new version is clearer and more accurate, and we look forward to your assessment of these latest modifications.
Reviewer: Maria Lazarova et al. study on Extracts of Sideritis scardica and Clinopodium vulgare Alleviate Cognitive Impairments in Scopolamine-Induced Rat Dementia is designed well and performed nicely, but still the manuscript needs some improvement and can be accepted after major revision. The comments and suggestions are given below:
Is the concentration mentioned by the author in the article the optimal therapeutic concentration? How do you determine which concentration to use? Are there other concentrations that have better results?
Authors’ Reply: Thank you for your question regarding the concentrations used in our study.
Our research primarily aimed to assess the therapeutic potential of S. scardica and C. vulgare, traditionally used in herbal remedies, for treating neurodegenerative diseases. This study represents an initial exploration in this field. The chosen doses for S. scardica (200 mg/kg) and C. vulgare (100 mg/kg), and their 1:1 combination, were determined based on existing literature references [113, 114] and prior empirical evidence.
In our experimental setup, we administered these extracts over a 21-day period, which included a pre-treatment phase of 10 days before and an 11-day treatment phase concurrent with Sco administration. The extracts were given one hour before each Sco injection to the rats. Our findings indicated that these doses were effective; but we recognize the importance of exploring broader concentration ranges. Moving forward, we plan to include additional concentrations and other in vivo neurodegenerative disease models in our research. This approach will enable a comprehensive examination of the dose-response relationship and may help identify more effective concentrations.
[113] Jeremic, I.; Petricevic, S.; Tadic, V.; Petrovic, D.; Tosic, J.; Stanojevic, Z.; Petronijevic, M.; Vidicevic, S.; Trajkovic, V.; Isakovic, A. Effects of Sideritis scardica extract on glucose tolerance, triglyceride levels and markers of oxidative stress in ovariectomized rats. Planta Med. 2019, 85(6), 465–472.
[114] Nasar-Eddin, G.; Simeonova, R.; Zheleva-Dimitrova, D.; Gevrenova, R.; Savov, I.; Bardarov, K.; Danchev, N. Beneficial effects of Clinopodium vulgare water extract on spontaneously hypertensive rats. Bul Chem Comm 2019, 51(A), 156–160.
Reviewer: The authors mentioned that Sco-induced downregulation of p-CREB/BDNF signaling demonstrated the protective effect of extracts. How can the neuroprotective effect be determined from the results presented that do not demonstrate upregulation of p-CREB/BDNF signaling after crude extract treatment? Relevant evidence should be provided that p-CREB/BDNF signalling is involved in neuroprotection.
Authors’ Reply: We acknowledge the Reviewer observation that the impact of the two extracts, as well as their combination, on p-CREB/BDNF signaling is not strictly stimulatory but rather involves mitigating the effects of scopolamine. In response to this point, we have revised the Discussion section to provide a clearer explanation of our findings and their implications. The updated content has been incorporated into the manuscript to address this concern (page 13 in the revised version).
Reviewer: This study found that C. vulgare S. scardica and their combination can improve learning and memory abilities, but there is no evidence of its potential mechanism of action. Some experiments are needed to prove its mechanism of action.
Authors’ Reply: Our research primarily aimed to assess the potential of S. scardica and C. vulgare, both commonly used in traditional remedies, for treating neurodegenerative diseases. It is important to note that this work should be considered a pilot investigation, showing the need for further detailed research to elucidate the exact mechanisms of action of these extracts. Despite this, our findings provide several important insights:
This combination of S. scardica and C. vulgare was notably effective, showing the best results in preserving recognition memory, robust antioxidant activity and superior preservation of monoaminergic function in the cortex. S. scardica showed remarkable preservation of spatial working memory. In comparison, C. vulgare not only matched S. scardica in preserving recognition memory but also exhibited the most potent antioxidant effect and the strongest acetylcholinesterase inhibitory activity. All plant extracts effectively counteracted the downregulation of p-CREB/BDNF signaling induced by scopolamine, an important finding in the context of neuroprotection. This study does not delve into the detailed mechanisms behind these effects, but it paves the way for further research into the potential therapeutic roles of these extracts in memory enhancement and neuroprotection.
Reviewer: The main components of S. scardica, C. vulgare, and their combination are phenolic substances. Which type of bioactive components play the main role? The neuroprotective effect should be further verified in vitro using hippocampal neuron cell models (such as HT22 cells).
Authors’ Reply: We are very thankful for the valuable suggestion to test the neuroprotective effects using specific hippocampal neuron cell models, like HT22 cells. This suggestion aligns perfectly with our goal to further understand the therapeutic potential of S. scardica and C. vulgare.
Regarding your question about the main active components:
Indeed phenolic compounds are believed to underlie the observed therapeutic effects of S. scardica and C. vulgare extracts. The largest portion of polyphenols in S. scardica consists of phenylpropanoids derivatives, including verbascoside and forsythoside A among many others [99]. It is known that forsythoside A possesses neuroprotective properties and can mitigate Alzheimer's-like pathology by inhibiting ferroptosis-mediated neuroinflammation [100]. Furthermore, clinical studies have shown the bioavailability of polyphenols found in S. scardica, with approximately 5% of ingested polyphenols from a cup of tea being detected as metabolites in urine samples [101]. The metabolite profiling of C. vulgare reveals that caffeic acid, chlorogenic acid and catechin are the most abundant secondary metabolites in the plant, and it seems that presence of a caffeoyl moiety in the structure of phenolic acids is essential for their anti-inflammatory activity [102]. This new information has been incorporated in the revised manuscript on page 13).
However, the further detailed analysis is needed to find out which specific components contribute to the neuroprotective effects.
We intend to implement your advice in our upcoming research, by using hippocampal neuron cell models. This approach will not only help us verify the protective effects of these extracts but also assist in identifying the critical active components.
[99] Petreska, J.; Stefkov, G.; Kulevanova, S.; Alipieva, K.; Bankova, V.; Stefova, M. Phenolic compounds of mountain tea from the Balkans: LC/DAD/ESI/MSn profile and content. Nat. Prod. Commun. 2011, 6(1), 21–30.
[100] Wang, C.; Chen, S.; Guo, H.; Jiang, H.; Liu, H.; Fu, H.; Wang, D. Forsythoside a mitigates Alzheimer's-like pathology by inhibiting ferroptosis-mediated neuroinflammation via Nrf2/GPX4 axis activation. Int J Biol Sci. 2022, 18(5), 2075–2090.
[101] Petreska Stanoeva, J.; Stefova, M. Assay of urinary excretion of polyphenols after ingestion of a cup of mountain tea (Sideritis scardica) measured by HPLC-DAD-ESI-MS/MS. J Agr Food Chem. 2013, 61(44), 10488–10497.
[102] Amirova, K.M.; Dimitrova, P; Marchev, A. S.; Aneva, I. Y.; Georgiev, M. I. Clinopodium vulgare L.(wild basil) extract and its active constituents modulate cyclooxygenase-2 expression in neutrophils. Food Chem. Toxicol. 2019, 124, 1–9.
Round 2
Reviewer 2 Report
Comments and Suggestions for Authors
OK
Reviewer 4 Report
Comments and Suggestions for Authors
The manuscript can be accepted in a present form.